# Furthest-Pair-Based Decision Trees: Experimental Results on Big Data Classification

**Ahmad B. A. Hassanat**

Information Technology College, Mutah University, Karak 61710, Jordan; hasanat@mutah.edu.jo;
Tel.: +962-79-889-7192

**Abstract:** Big Data classification has recently received a great deal of attention due to the main properties of Big Data, which are volume, variety, and velocity. The furthest-pair-based binary search tree (FPBST) shows a great potential for Big Data classification. This work attempts to improve the performance the FPBST in terms of computation time, space consumed and accuracy. The major enhancement of the FPBST includes converting the resultant BST to a decision tree, in order to remove the need for the slow K-nearest neighbors (KNN), and to obtain a smaller tree, which is useful for memory usage, speeding both training and testing phases and increasing the classification accuracy. The proposed decision trees are based on calculating the probabilities of each class at each node using various methods; these probabilities are then used by the testing phase to classify an unseen example. The experimental results on some (small, intermediate and big) machine learning datasets show the efficiency of the proposed methods, in terms of space, speed and accuracy compared to the FPBST, which shows a great potential for further enhancements of the proposed methods to be used in practice.

**Keywords:** machine learning; AI; BST; diameter; algorithms; d-dimensional datasets; decision tree

## 1. Introduction

Big Data analytics has received a great deal of attention recently, particularly in terms of classification, this is due to the main properties of Big Data: volume, variety, and velocity [1,2]. Having a large number of examples and various types of data, Big Data classification attempts to seize these properties to obtain better learning models with fast learning/classification [3–5].

The problem of Big Data classification is similar to the tradition classification problem, taking into consideration the main properties of such data, and can be defined as follows, given a training dataset of n examples, in d dimensions or features; the learning algorithm needs to learn a model that will be able to efficiently classify an unseen example E. In the case of Big Data, where n and/or d are very large values, tradition classifiers become inefficient, for example the K-nearest neighbors (KNN) [6,7] took weeks to classify some Big Data sets [8].

Recently, we proposed three methods for Big Data classification [8–10]. All of these methods employ an approach based on creating a binary search tree (BST), in order to speed up the Big Data classification using a KNN classifier with a smaller number of examples, those which are found by the search process. The real distinction between these methods is in the way of creating the BST. The first uses the furthest-pair of points to classify the examples along the BST, the second uses two extreme points based on the minimum and maximum points found in a dataset, and the third uses the Euclidean norms of the examples. Each has its own weakness and strength. However, the common weakness is the use of the slow KNN classifier.

The main goal and contribution of this paper is to improve the performance of the first method-the furthest-pair-based BST (FPBST), by removing the need for the slow KNN classifier, and converting

the BST to a decision tree (DT). However, any enhancement made for this method can be easily generalized to the other two methods.

The new enhancement might make the FPBST (and its sisters) a practical alternative for the KNN classifier, since the KNN might be the only available choice in certain cases, for example, such as when used for content-based image retrieval (CBIR) [11,12].

The FPBST sorts the numeric training examples into a binary search tree, to be used later by the KNN classifier, attempting to speed up the Big Data classification, knowing that searching a BST for an example is much faster than searching the whole training data. This method depends mainly on finding two local examples (points) to create the BST, these points are the furthest-pair of points (diameter) of a set of points in d-dimensional Euclidean feature space [9], these two points are found using a greedy algorithm proposed by [13]. These points are then used to sort the other examples based on their similarity using the Euclidean distance. The training phase of the FPBST ends by creating the BST, which is searched later for a test example E to a leaf-node, where similar examples are found, the KNN classifier is then used to classify E.

Having known that the KNN is slow, we opt for disusing it in this paper, and we do this by converting the resultant BST to a decision tree. To do so, we opt for calculating the probabilities of each class at each node, we calculate the probabilities using four methods, (1) calculating them at the leaf node only; (2) calculating the accumulated probabilities along the depth of the tree; (3) calculating the weighted-accumulated probabilities using the tree's level as a weight; and (4) calculating the weighted-accumulated probabilities using the tree's level as an exponential weight. Therefore, we propose four methods based on these four calculations, these four methods stop clustering when having examples of only one class. We propose the fifth method which uses the accumulated probabilities of the classes but continues clustering until there is only one example (or similar examples) in a leaf-node.

We further enhanced these five methods by swapping the furthest-pair of points based on the minimum class, so as to obtain a coherent decision tree, where examples of similar classes are stored closer to each other, unlike the FPBST, which uses the minimum/maximum norms for this purpose, thus, we propose ten methods in this paper. These methods/enhancements of the FPBST solve (by default) another related problem associated with the FPBST use of the KNN, which is finding the best k for the KNN [14,15]. In this work, there is no need to determine such a parameter since there is no need to use the KNN.

The important of this research stems from the decreased size of the resultant tree, which is attained by trimming the tree, where all the examples found in a node were of the same class, and this speeds up the training process, reduces the space needed for the resultant tree and increases the speed of testing, in addition to increasing the accuracy of classification if possible.

The rest of this paper is organized as follows. Section 2 presents some related methods used for Big Data classification. Section 3 describes the proposed enhancements and the data set used for experiments. Section 4 evaluates and compares the proposed enhancements to FPBST. Section 5 draws some conclusions, shows the limitations of the proposed enhancements, and gives directions for future research.

## 2. Related Work

Recently, remarkable efforts have been made to find new methods for Big Data classification, in addition to the FPBST, reference [10] used two extreme points, which are the minimum and maximum points found in a dataset to create a BST, so as to sort the examples of a training set. This BST is then searched for a test example to a leaf-node, where similar examples can be found, the KNN classifier is then used to classify a test example. Similarly, reference [8] used the same methodology, except for the manner of creating the BST, where it was created based on the Euclidean norms of the training examples. Both methods were very fast, however, the accuracy results were slightly less than that of the FPBST [9] in general.

Two recent and interesting approaches proposed by Wang et al. [16] deal with the problem differently, using random and principal component analysis (PCA) techniques to divide the data in order to obtain multivariate decision tree classifiers. Both methods were evaluated on several Big Datasets, the reported accuracy results considering all the datasets used, show that the data partitioning using PCA performs better than that of a random technique used.

Maillo et al. [17] proposed a parallel implementation based on mapping the training set examples, followed by reducing the number of examples that are related to a test sample. The reported results were similar to that of an exact KNN but faster, i.e., about up to 149 times faster than the KNN when tested on 1 million examples; the speed of this parametric method depends mainly on the K neighbors as well as the number of maps used. This work is further improved by almost the same team [18], where they proposed a new KNN based on Spark, which is similar to the mapping/reducing technique but with using multiple reducers to speed up the process, the size of the dataset used was up to 11 million examples.

Based on clustering the training set using K-means clustering algorithm, Deng et al. [19] proposed two methods to increase the speed of KNN, the first used random clustering and the second used landmark spectral clustering, when finding the related cluster, both utilize the KNN to test the input example with a smaller set of examples. Both algorithms were evaluated on nine Big Datasets showing reasonable approximations to the sequential KNN, the reported accuracy results were dependent on the number of clusters used.

Another clustering approach is utilized recently by Gallego et al. [20], who proposed two clustering methods to accelerate the speed of the KNN, both are similar; however, the second is an enhancement of the first, where a cluster augmentation process is employed. The reported average accuracy of all the Big Datasets used was in the range of 83 to 90% depending on the K-neighbors and number of clusters used. The performance of both methods has improved significantly when the Deep Neural Networks has been employed for learning a suitable representation for the classification task.

Most of the proposed work in this domain is based on divide and conquer approach, this is a logical approach to use with Big Datasets and, therefore, most of these approaches are based on clustering, splitting, or partitioning the data to turn and reduce the very large size to a manageable size that can be used for and efficient classification. One major problem associated with such approaches is that the determination of the best number of clusters/parts, sine more clusters means fewer examples and, therefore, faster testing. However, fewer examples also means less accuracy, as the examples found in a specific cluster might not be related to the tested example. On the contrary, few clusters indicate a large number of examples per each, which increases the accuracy but slows down the classification process if the KNN is used.

Similar to [8–10] there exist extensive literature on tree structures such as k-d trees [21], metric trees [22], cover trees [23], and other related work such as [24,25]. Regardless of the plethora of the proposed methods in this domain, there is still room for improvement in terms of accuracy and time consumed for both training and testing stages. Additionally, this work is nothing but an attempt to improve the performance of one of these methods.

## 3. Furthest-Pair-Based Decision Trees (FPDT)

This section describes and illustrates the proposed methods, in addition to describing the data used for evaluation and experiments.

### 3.1. Methods

The main improvement of the FPBST [9] includes the unemployment of the standard KNN algorithm as described by [6,7], which is time-consuming particularly when classifying Big Data. In this paper, we propose the use of the probabilities of the classes found in the leaf-nodes to decide the class of a test example, without having to use the slow KNN, even if there are a small number of examples found in a leaf-node. We keep the same functionality of the binary search tree (BST), which

is employed to sort the examples (points) of machine learning datasets in a way that facilitates the search process. This BST sorts all the examples taken from a training dataset based on their distances from two local points (P1 and P2), which are two examples from the training dataset itself, and they vary based on the host node and the level/location of that node in a BST.

The FPBST builds its BST by finding the furthest points P1 and P2 [13], assuming that the furthest points are the most dissimilar points, and therefore, are more likely to be belonging to different classes. Thus, sorting other examples based on their distances to these points might be a good choice, as similar examples are sorted nearby, while dissimilar examples are sorted faraway in the created BST.

Similar to the FPBST, the training phase of the proposed method (FPDT) creates a binary decision tree (DT), which speeds up searching for a test example comparing to the unacceptable time an exhaustive search, particularly when classifying Big Datasets. We use the same Euclidean distance metric (ED) for measuring distance, to compare the results of the proposed method to those of the FPBST.

While creating the DT, we calculate the probability of each class to occur in each node, her we opt for several options:

1.  Accumulate the classes' probabilities by adding the parent's probabilities to its children's; we call this method decision tree 0 (DT0).
2.  Accumulate the probabilities by adding the parent's probabilities to its children's; and weighting these probabilities by the level of the node, assuming that the more we go deeper in the tree, the more likely we reach to a similar example(s), this is done by multiplying the tree level by the classes' probabilities at a particular node; we call this method decision tree 1 (DT1), and is shown in Algorithm 1 as (Dtype = 1).
3.  Accumulate the classes' probabilities by adding the parent's probabilities to its children's; and weighting these probabilities exponentially, for the same reason in 2, but with higher weight. This is done by multiplying the classes' probabilities by 2 to the power of the tree's level at a particular node; we call this method decision tree 2 (DT2), and is shown in Algorithm 1 as (Dtype = 2).
4.  No accumulation of the probabilities, we use just the probabilities found in a leaf node; we call this method decision tree 3 (DT3), and is shown in Algorithm 1 as (Dtype $\neq$ 3).
5.  Similar to DT0 (normal accumulation), but the algorithm continues to cluster until there is only one or a number of similar examples in a leaf-node, this is done even if all the examples of a current node belong to the same class. While DT0–DT3 stop the recursive clustering when all the examples of the current node are belonging to the same class, and consider the current class as a leaf-node, we call this method decision tree 4 (DT4), and is shown in Algorithm 1 as (Dtype = 4).

The idea behind accumulating the probabilities is to remove the effect of unbalanced datasets, as some datasets contains more examples of a specific class than the other classes, and this will increase the probability of the dominant class, since it is calculated in the root node and accumulated along the depth of the tree, so by moving deeper, less number of the dominant examples remain.

Algorithm 1 shows the pseudo code for the training phase of the FPDT method, which works well for DT0, DT1, DT2, DT3, and DT4 depending on the input (Dtype), and Algorithm 2 shows the pseudo code for the testing phase of the FPDT method, which is the same for DT0, DT1, DT2, DT3, and DT4, as these methods differ in the way of creating the decision tree only, i.e., the training phase.

---

**Algorithm 1.** Training Phase (DT building) of FPDT.

---

**Input:** Numerical training dataset DATA with *n* FVs and *d* features, and DT type (Dtype)
**Output:** A root pointer (RootN) to the resultant DT.
1- Create a DT Node → RootN
2- RootN.Examples ← FVs//all indexes of FVs from the training set
3- (P1, P2) ← **Procedure** Furthest(*DATA* ← RootN.Examples, n)//hill climbing algorithm [1]
4- **if** EN(P1) > EN(P2) swap(P1, P2)
5- RootN.P1 ← P1
6- RootN.P2 ← P2
7- RootN.Left = Null
8- RootN.Right = Null
9- **Procedure** BuildDT(Node ← RootN)
10- **for** *each $FV_i$ in Node*, **do**
11-     D1←ED($FV_i$, Node.P1)
12-     D2←ED($FV_i$, Node.P2)
13-     **If** (D1 < D2)
14-         Add index of $FV_i$ *to* Node.Left.Examples
15-     **else**
16-         Add index of $FV_i$ *to* Node.Right.Examples
17- **end for**
18- **if** (Node.Left.Size == 0 **or** Node.Right.Size == 0)
19-     **return** //this means a leaf node
20- (P1, P2) ← Furthest(Node.Left.Examples, size(Node.Left.Examples))//*work on the left child*
21- **if** (EN(P1) > EN(P2)) then swap(P1, P2)
22- Node.Left.P1 ← P1
23- Node.Left.P2 ← P2
24- Node.Left.ClassP [numclasses] = {0}//*initialize the classes' probabilities to 0;*
25- **for** *each i in* Node.Left.Examples **do**
26-     Node.Left.ClassP [DATA.Class[i]]++//*histogram of classes at Left-Node*
27- bool    LeftMulticlasses = false//check for single class to prune the tree
28- **if** *there is* **more** *than one class at* Node.Left.ClassP
29-     LeftMulticlasses=true;
30- if (Dtype ==4)    //*no pruning if chosen*
31-     LeftMulticlasses=true//*even if there is only one class in a node=> cluster it further*
32- **for** *each i in* numclasses **do** //*calculate probabilities of classes at the left node*
33-     Node.Left.ClassP [i]= Node.Left.ClassP [i]/ size(Node.Left.Examples)
34- **if** (Dtype ==1) //*increase the probabilities by the increased level*
35-     **for** *each i in* numclasses **do**
36-         Node.Left.ClassP [i]= Node.Left.ClassP [i]* Node.Left.level
37- **if** (Dtype ==2) //*increase the probabilities exponentially by the increased level*
38-     **for** *each i in* numclasses **do**
39-         Node.Left.ClassP [i]= Node.Left.ClassP [i]* $2^{Node.Left.level}$
40- **if** (Dtype != 3)//*do accumulation for probabilities, if 3, use just the probabilities in a leaf node*
41-     **for** *each i in* numclasses **do**
42-         Node.Left.ClassP [i] = Node.Left.ClassP [i] + Node.ClassP [i]
43- Node.Left.Left = NULL;
44- Node.Left.Right = NULL;
45- **Repeat** the previous steps (20–44) on Node.Right
46- **if** (LeftMulticlasses)
47-     BuildDT (Node.Left)
48- **if** (RightMulticlasses)
49-     BuildTree(Node.Right)
50- **end Procedure**
51- **return** RootN
52- **end** Algorithm 1

---

---

**Algorithm 2.** Testing Phase of FPDT.

---

**Input:** test dataset TESTDATA with *n* FVs and *d* features
**Output:** Testing Accuracy (Acc).
1- Acc←0
2- **for** *each $FV_i$ in* TESTDATA **do**
3-          **Procedure** GetTreeNode(Node ← RootN, $FV_i$)
4-                 D1 ← ED(FV[*i*], Node.P1)
5-                 D2 ← ED(FV[*i*], Node.P2)
6-                 **if** (D1 < D2 **and** Node.Left)
7-                        **return** GetTreeNode (Node.Left, $FV_i$)
8-                 **else if** (D2 ≤ D1 **and** Node.Right**)**
9-                        **return** GetTreeNode (Node.Right,$FV_i$)
10-                  **else**
11-                         **return** Node
12-                 **end if**
13-         **end Procedure** GetTreeNode
14-         class ← argmax(Node.ClassP)// *returns the class with the maximum probability*
15-         **if** class == Class($FV_i$)
16-                Acc ← Acc+1
17- **end for** *each*
18- Acc← Acc/n
19- **return** Acc
20- end Algorithm 2

---

The training phase of the FPBST and the new DT0–DT4 use the Euclidean norm to regularize the resultant tree by swapping P1 and P2 if the norm of P2 is less than that of P1 (step 21 in Algorithm 1). This is normally done to let the examples, which are similar to the point of the least norm to be sorted to the left side of the tree, and the others to be sorted to the right side of the tree, so as to have similar examples adjacent as possible as could in the resultant BST. Having known that the Euclidean norm is sensitive to the negative numbers (negative and positive similar numbers result the same Euclidean norm), the examples with many zeros or similar repeated numbers [2], we opt for an alternative of the norm to decide which goes to left and which goes to right. Here we propose the use of the class of the example, so we check the classes of P1 and P2 to see if P2 has the minimum class, if yes, we swap P1 with P2, otherwise they remain as they are. Such a swap allows for regularizing the resultant decision tree with the minimum cost, as creating the norm cost extra O(d) each time, while obtaining the class of an example costs O(1), and at the same time we get more coherent trees in terms of the classes distribution, since the examples of minimum class are forced to be sorted to the left and those with the maximum class are sorted to the right, this might have a good effect on the probabilities of the classes. This improvement is applied on all the proposed DT0–DT4 making new decision trees DT0+, DT1+, DT2+, DT3+, and DT4+.

Similar to the FPBST, the time complexity of training phase to build the decision tree (DT) by the proposed methods (DT0–DT4 and DT0+ to DT4+) is:

$$T(n, d) = O(cnd \ \log \ n) \tag{1}$$

where (cnd) is the time consumed to find the approximate furthest points, as the constant c is the number of iterations needed to find the approximate furthest points, which is found experimentally to be in the range of 2 to 5 [1]. The (log n) time is consumed along the depth of the DT.

An extra (2nd) time is consumed by comparing each example or feature vector (FV) to the local furthest points (P1 and P2). This time can be added to c to make it in the range of 4 to 7, however, c is still a constant and the overall time complexity can be asymptotically approximated to:

$$T(n, d) = O(nd \ \log \ n) \tag{2}$$

and if n >> d, the time complexity can be further approximated to:

$$T(n, d) = O(n \ \log \ n) \tag{3}$$

The space complexity can be defined by:

$$S(n, d) = O(n \ \log \ n) \tag{4}$$

where the space consumed (S) is a function of n and d, which similar to the size of a normal BST.

The test phase of the proposed method (Algorithm 2) is the same for all the DTs, as it searches the created DT for a test example starting from the root node to a leaf node, where similar example(s) are supposed to be there. However, it is different from the test phase algorithm of the FPBST, where KNN algorithm is employed to classify the test example using those found in a leaf-node. The proposed DTs have no need to use the KNN, because the leaf-node has become able to decide the class of the tested example based on the pre-calculated probabilities it has, since the name (decision tree) suggests. Disusing the KNN with the proposed DTs allows for more speed. Therefore, the time complexity of the test phase of the proposed DTs for each tested example is:

$$T(n, d) = O(2d \ \log \ n) \tag{5}$$

where the (2d) time is consumed by the calculation of the ED, which costs d time for each comparison with either P1 or P2. And the (log n) time is consumed along the depth of the BST, which is about (log n) on average.

And if n >> d, the d time can be ignored making the testing time:

$$T(n, d) = O(\log \ n) \tag{6}$$

*3.2. Implementation Example*

To further explain the proposed Dts, we implement some of them to create decision trees to be compared with the BST of the FPBST. For this end, we used a small synthesized dataset for illustration purposes. The synthesized dataset used consists of two hypothetical features (X1 and X2) and two classes (0 and 1) having 20 examples as shown in Table 1 and illustrated in Figure 1.

**Table 1.** A hypothetical training data sample to exemplify the resultant BST of the FPBST, as well as the decision trees of the proposed methods.

| #Example | X1 | X2 | Class | Euclidean Norms |
|:---:|:---:|:---:|:---:|:---:|
| 0 | 4 | 3 | 0 | 5.0 |
| 1 | 2 | 5 | 0 | 5.4 |
| 2 | 2 | 4 | 0 | 4.5 |
| 3 | 4 | 4 | 0 | 5.7 |
| 4 | 3 | 6 | 0 | 6.7 |
| 5 | 1 | 0 | 0 | 1.0 |
| 6 | 1 | 3 | 0 | 3.2 |
| 7 | 3 | 1 | 0 | 3.2 |
| 8 | 3 | 2 | 0 | 3.6 |
| 9 | 4 | 6 | 0 | 7.2 |

**Table 1.** *Cont.*

| #Example | X1 | X2 | Class | Euclidean Norms |
|----------|----|----|-------|-----------------|
| 10 | 4 | 5 | 1 | 6.4 |
| 11 | 3 | 7 | 1 | 7.6 |
| 12 | 8 | 6 | 1 | 10.0 |
| 13 | 9 | 7 | 1 | 11.4 |
| 14 | 3 | 4 | 1 | 5.0 |
| 15 | 5 | 7 | 1 | 8.6 |
| 16 | 8 | 3 | 1 | 8.5 |
| 17 | 3 | 5 | 1 | 5.8 |
| 18 | 4 | 8 | 1 | 8.9 |
| 19 | 8 | 8 | 1 | 11.3 |

If we apply the FPBST on the synthesized dataset we get the BST illustrated in Figure 2, and when applying the DT0, DT0+, DT1, and DT1+ on the same dataset we get the decision trees illustrated in Figures 3–6, respectively.

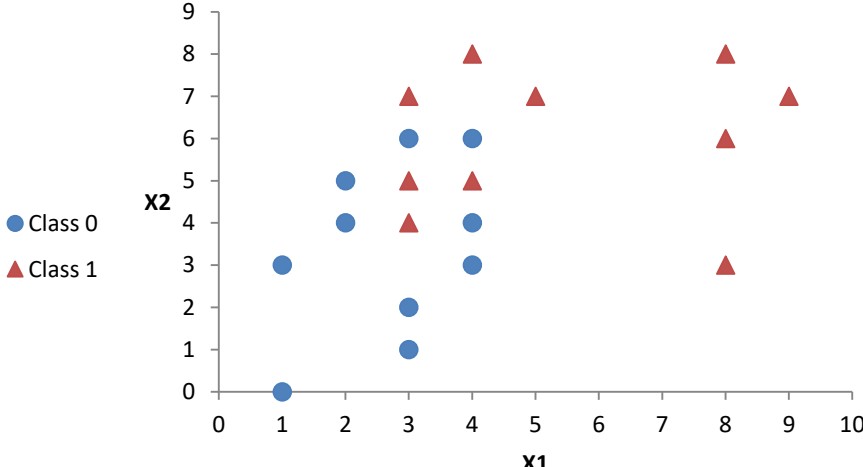

**Figure 1.** A visual illustration of the synthesized dataset obtained from Table 1.

The purpose of these figures is not to prove anything, as we cannot draw significant conclusions from such weak evidence (the very small data set in Table 1). However, they are meant to show how the proposed DTs are constructed comparing to the BST of the FPBST. It is interesting to note that calculating the furthest-pair of points is an approximate algorithm and might not obtain the same pair of points always, as seen in the Figures 2 and 6, where the furthest pair was (5, 19), while the furthest pair was (5, 13) in Figures 3–5. Both of the pairs have the maximum distance in the dataset which is 10.63. In addition, we can note the smaller size of the DTs in (Figures 3–6) and the shallow nodes comparing to the BST in Figure 2, this is because the DTs stop the recursive process to create child-nodes when the node is pure, i.e., all the examples hosted belong to the same class. One exception is the DT4 and DT4+, which carry on sorting the examples until there is only one example (or similar examples) in a leaf-node, we mean by similar examples, those who share the same Euclidean distance to a reference point. Additionally, we can note the difference between the DTs and the DT+s, for example, the point 14 is classified as class 0 in Figure 3, while it belongs to class 1, this is because its norm = 5, while the other point (1) sharing the same parent node has a norm = 5.4, according to DT0, Point 14 goes to the left and Point 1 goes to the right, if the DT0 was not calculating accumulated probabilities this should not make a big difference, but since such type of probabilities is used by the DT0 and DT0+ without giving a higher weight to the deeper levels we get such a classification error. However, this situation is not happening when using DT1 and DT1+, because the tree level is used to weight the probabilities.

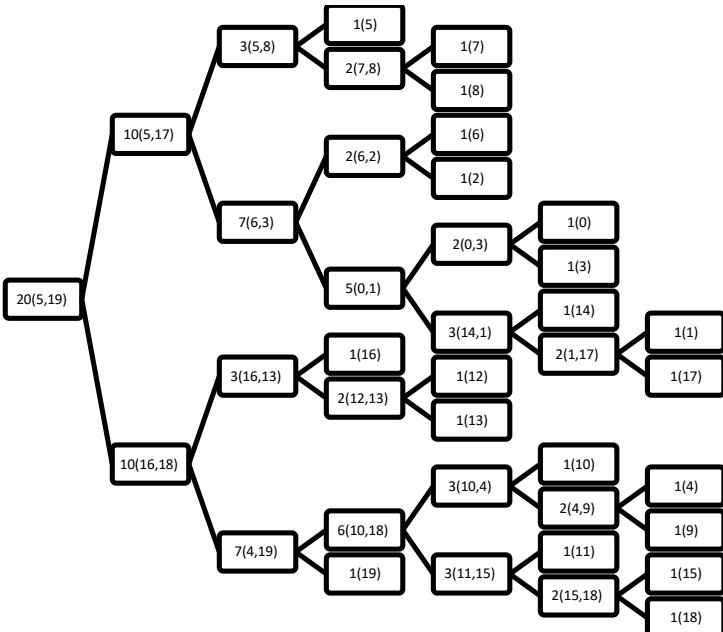

**Figure 2.** The resultant BST after applying the training phase of the FPBST on the sample data from Table 1. The number outside the brackets is the counter of the examples hosted by each node, and those inside the brackets are the index of the examples in a leaf node, or the furthest points (P1 and P2) otherwise.

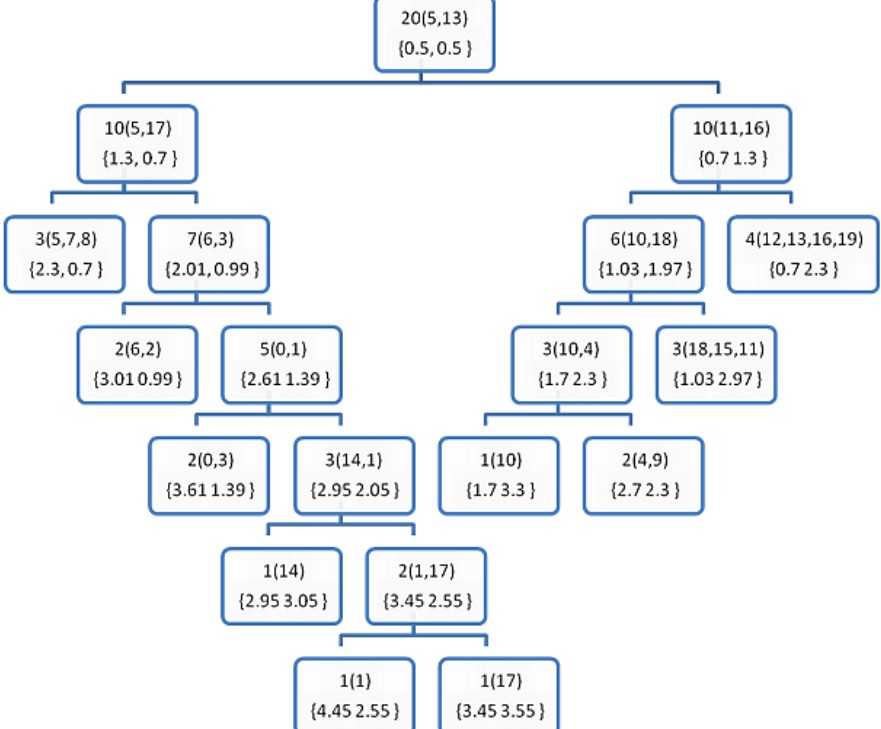

**Figure 3.** The resultant decision tree after applying the training phase of the DT0 on the sample data from Table 1. The number outside the rounded brackets () is the counter of the examples hosted by each node, and those inside the rounded brackets are the index of the examples in a leaf node, or the furthest points (P1 and P2) otherwise. The numbers in the curly brackets {} shows the probabilities of the classes at each node.

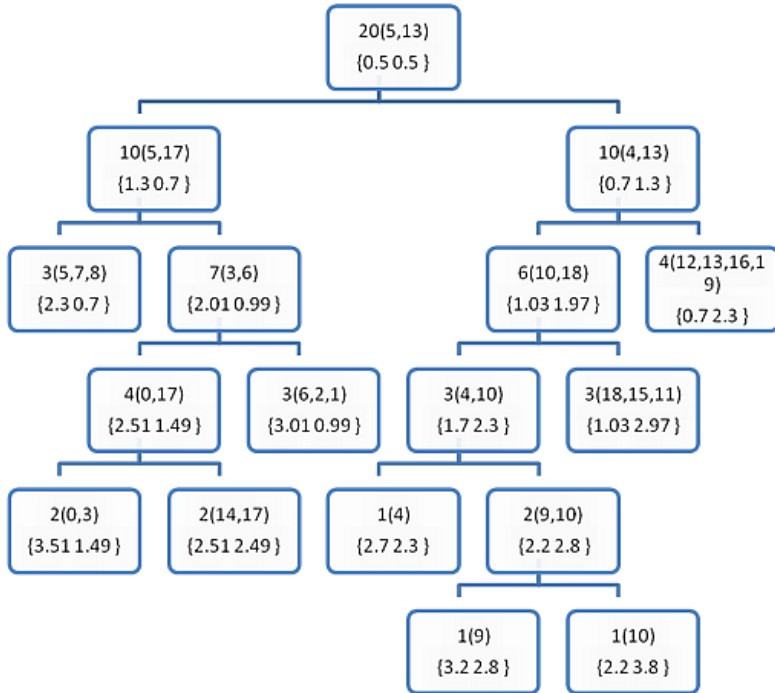

**Figure 4.** The resultant decision tree after applying the training phase of the DT0+ on the sample data from Table 1. The number outside the rounded brackets is the counter of the examples hosted by each node, and those inside the rounded brackets are the index of the examples in a leaf node, or the furthest points (P1 and P2) otherwise. The numbers in the curly brackets {} shows the probabilities of the classes at each node.

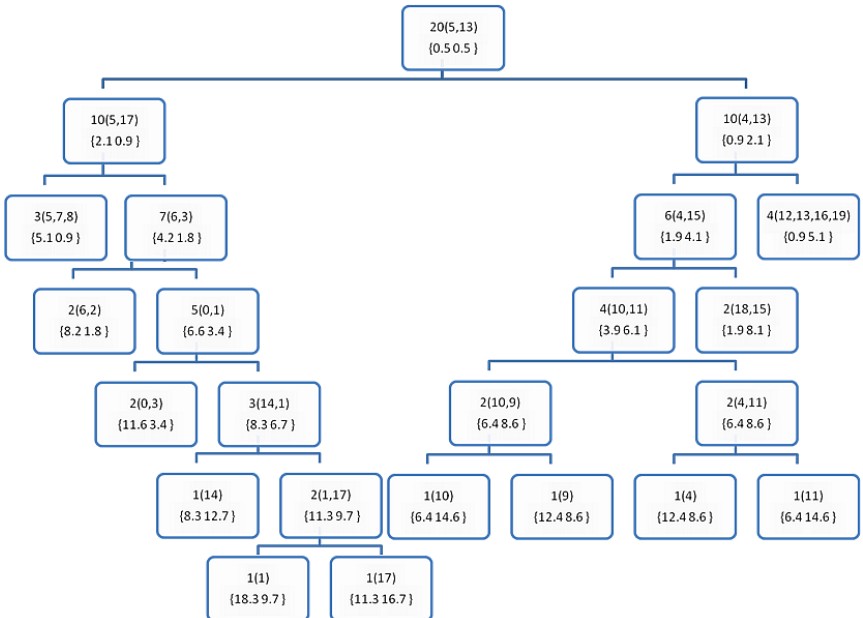

**Figure 5.** The resultant decision tree after applying the training phase of the DT1 on the sample data from Table 1. The number outside the rounded brackets is the counter of the examples hosted by each node, and those inside the rounded brackets are the index of the examples in a leaf node, or the furthest points (P1 and P2) otherwise. The numbers in the curly brackets {} shows the probabilities of the classes at each node.

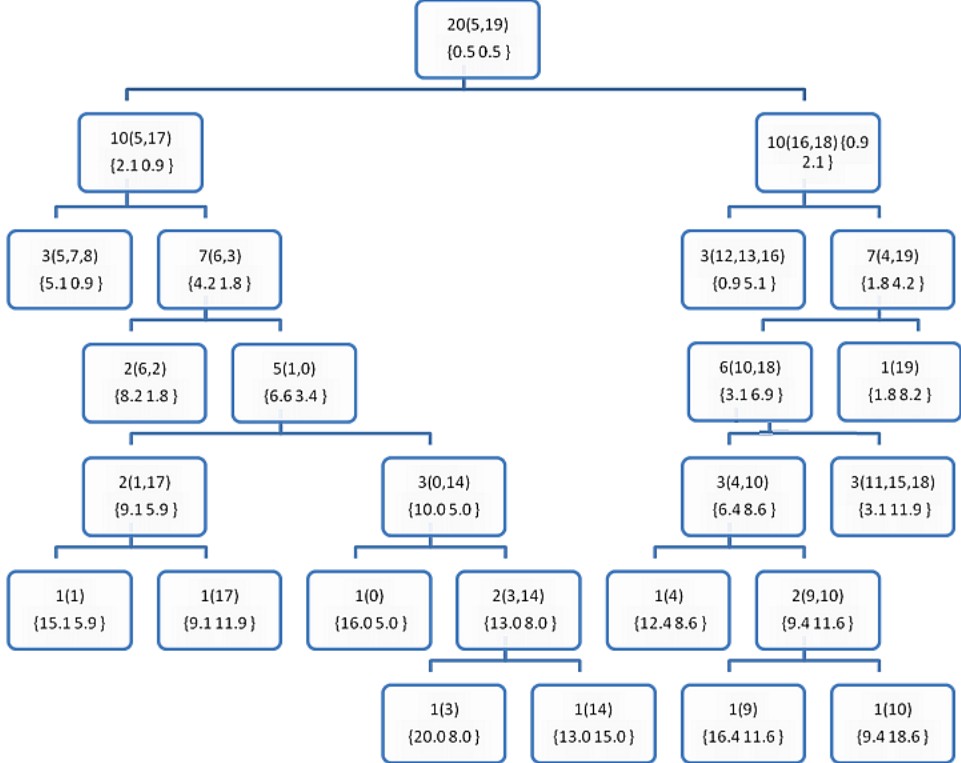

**Figure 6.** The resultant decision tree after applying the training phase of the DT1+ on the sample data from Table 1. The number outside the rounded brackets is the counter of the examples hosted by each node, and those inside the rounded brackets are the index of the examples in a leaf node, or the furthest points (P1 and P2) otherwise. The numbers in the curly brackets {} shows the probabilities of the classes at each node.

*3.3. Data*

In order to evaluate the proposed methods and compare the results to the FPBST on Big Data classification, we use some of the well-known machine learning datasets, which are used by state-of-the-art work in this domain. These datasets are freely available for download from either the support vector machines library (LIBSVM) Data [26] or the UCI Machine Learning Repository [27]. The datasets used are of different dimensions, sizes, and data types, such diversity is important to evaluate the efficiency of the proposed method in terms of accuracy and time consumed.

All datasets used contain numeric data, i.e., real numbers and/or Integers. The sizes of these datasets are in the range of 625 to 11,000,000 examples; the dimensions are in the range of 4 to 5000 features. Table 2 shows the descriptions of the datasets used.

**Table 2.** Description of datasets used for evaluation and comparison of the proposed methods.

| Dataset | Size | Dimensions | Type | #Class |
|---------|------|------------|------|--------|
| HIGGS | 11,000,000 | 28 | Real | 2 |
| SUSY | 5,000,000 | 18 | Real | 2 |
| Poker | 1,025,010 | 11 | Integers | 10 |
| Covtype | 581,012 | 54 | Integers | 7 |
| Mnist | 70,000 | 784 | Integers | 10 |
| Connect4 | 67,557 | 42 | Integers | 3 |
| Nist | 44,951 | 1024 | Integers | 26 |
| LetRec | 20,000 | 16 | Real | 26 |
| Homus | 15,199 | 1600 | Integers | 32 |
| Gisette | 13,500 | 5000 | Integers | 2 |

| Dataset | Size | Dimensions | Type | #Class |
|---------|------|------------|------|--------|
| Pendigits | 10,992 | 16 | Integers | 10 |
| Usps | 9298 | 256 | Real | 10 |
| Satimage | 6435 | 36 | Real | 6 |
| Abalone | 4177 | 8 | Real | 3 |
| Climate | 1178 | 18 | Real | 2 |
| German | 1000 | 24 | Integers | 2 |
| Blood | 748 | 4 | Integers | 2 |
| Australian | 690 | 14 | Real | 2 |
| Cancer | 683 | 9 | Integers | 2 |
| Balance | 625 | 4 | Integers | 3 |

## 4. Results and Discussion

To evaluate the proposed methods (DT0–DT4 and DT0+-DT4+), we programmed both Algorithms 1 and 2 using MS VC++.Net framework, version 2017, and conducted several classification experiments on all the datasets described in the data section. We utilized a personal computer with the following specifications:

- Processor: Intel®Core™ i7-6700 CPU @ 340GHz
- Installed memory (RAM): 16.0 GB
- System type: 64-bit operating system, x64-based processor, MS Windows 10.

Table 3 shows the characteristics of the BT built using the proposed DTs comparing to that of the FPBST, her we used one dataset (poker), as being one of the largest datasets and to save space for this paper.

**Table 3.** Some specifications of the resultant BST of the FPBST compared to the resultant DTs after applying the proposed FPDTs on the poker dataset (training phase).

| Method | Number of Nodes | Number of Leaves | Maximum Depth | Total Examples in All Leaves | Min Number of Examples in a Leaf | Max Number of Examples in a Leaf |
|--------|-----------------|------------------|---------------|------------------------------|----------------------------------|----------------------------------|
| FPBST | 2,045,541 | 1,022,771 | 30 | 1,025,010 | 1 | 3 |
| DT0 | 1,433,617 | 716,809 | 29 | 1,025,010 | 1 | 49 |
| DT1 | 1,433,089 | 716,545 | 29 | 1,025,010 | 1 | 80 |
| DT2 | 1,433,631 | 716,816 | 29 | 1,025,010 | 1 | 71 |
| DT3 | 1,432,131 | 716,066 | 30 | 1,025,010 | 1 | 47 |
| DT4 | 2,045,541 | 1,022,771 | 30 | 1,025,010 | 1 | 3 |
| DT0+ | 1,440,047 | 720,024 | 29 | 1,025,010 | 1 | 99 |
| DT1+ | 1,439,295 | 719,648 | 29 | 1,025,010 | 1 | 42 |
| DT2+ | 1,439,113 | 719,557 | 30 | 1,025,010 | 1 | 56 |
| DT3+ | 1,441,107 | 720,554 | 30 | 1,025,010 | 1 | 44 |
| DT4+ | 2,045,541 | 1,022,771 | 30 | 1,025,010 | 1 | 3 |

As can be noted in Table 3, the maximum depth of the resultant BST and DTs is not much larger than log2(1025010) = 19.97, this of course increases the speed of the test phase for all the proposed methods including the FPBST. Although the number of nodes in a full BST is typically (n log n), and therefore should be around 20,421,879, we found it much less than that for all methods, this is due to the resultant BST and DTs being not full binary trees. It is interested to note that the number of nodes in the proposed DTs is significantly less than that of the BST; this is related to the number of hosted examples in the leaf-nodes, as it is higher in the DTs than the BST, i.e., the lower the number of nodes, the higher the number of examples per leaf-node. This is because the DTs stop the recursive process earlier, mainly, when all the existing examples are belonging to only one specific class. One exception is the DT4 and DT4+, obviously because both of them do not stop the recursive process and carry on creating nodes until there is only one example per each leaf-node, or similar examples.

The relatively small size of the DT created by the proposed DT0–DT3 and DT0+- DT3+ shall serve two purposes, (1) decreasing the space needed for the tree; and (2) speeding up the classification process, since searching a smaller tree is faster than a larger one. This is also complying with the number of leaf-node, as being significantly smaller than that of the FPBST, DT4, and DT4+.

In this paper, we compare the performance of the proposed methods to that of the FPBST, as the goal of this paper is to improve the performance of the FPBST, in terms of speed, space used, and classification accuracy. For this end, we evaluated the proposed methods DT0–DT4 by employing them to classify the machine learning datasets stated in Table 2, using 10-fold cross-validation, so as to be able to compare their performances to that of the FPBS.

Ideally, the 10-fold cross-validation approach selects the training data set randomly; however, Nalepa and Kawulok [28] discussed other interesting methods for selecting the training data such as data geometry analysis (clustering and non-clustering), neighborhood analysis methods, Evolutionary, active learning and the random sampling methods. Our choice belongs to the random sampling methods as being the most used.

Since we used a different hardware with different computation powers, which might significantly affect the comparison in terms of time consumed, we opt for reporting the speed-up factor of each method similarly to [13,17]. We calculate the speed-up factor by considering the ratio of the time consumed by the FPBST classifier to that of the proposed methods on the same dataset used and same examples tested as follows:

$$\text{Speedup}(X, D) = \frac{T(\text{FPBST}, D)}{T(X, D)} \tag{7}$$

where D is the dataset tested, X is the method that we wish to calculate its speedup factor, and T is the time function, which returns the time consumed by the method X on the dataset D.

The accuracy comparison results are shown in Table 4. Tables 5 and 6 show the time consumed in the training and testing phases, respectively, while Table 7 shows the speed-up comparison results.

As can be seen from Table 4, one or more of the proposed methods DT0–DT4 slightly outperform the FPBST in terms of accuracy when testing on all datasets except for the Satimage, which works better with the FPBST, however, the difference is not significant (less than 1%), and it might due to randomness of the train/test examples, on average, we can see that the DT1, DT3, and DT4 perform slightly better than the FPBST. The maximum average classification accuracy is attributed to the DT4; this is due to the nature of the DT that created by the DT4, which continues the recursive process until there is only one class or similar classes per leaf-node. We are not favoring the DT4 as its size is similar to the BST of the FPBST, however, its accuracy is not significantly higher than the other DTs and the FPBST, for example the DT3 outperforms all methods in terms of the number of datasets tested. We can say with some confidence that the proposed approach (using the decision tree instead of the KNN-regardless the creation method of the decision tree) performs well on all the evaluated datasets, and this performance is almost similar to the FPBST in some cases or slightly better in other cases.

It is interesting to note from Table 5 and Figure 7 that the proposed DT0–DT3 consumed less time in general than the FPBST and the DT4, this is due to the smaller decision trees created by these methods, However, the time saved while building the decision tree by DT0–DT3 is not significant on some datasets, this is due to the extra calculations of the probabilities of each class for each dataset. It is also interesting to note the time consumed by the DT4 is almost similar to that of the FPBST and sometimes longer; this is because it has a similar tree size to that of the FPBST, with extra time for calculating the probabilities.

**Table 4.** Accuracy results of the proposed methods DT0–DT4 compared to that of the FPBST, using 10-fold cross-validation.

| Dataset | FPBST | DT0 | DT1 | DT2 | DT3 | DT4 |
|---|---|---|---|---|---|---|
| Abalone | 0.4990 | **0.5374** | 0.5338 | 0.4906 | 0.5122 | 0.5326 |
| Australian | 0.6435 | 0.6667 | 0.6725 | 0.6203 | 0.6392 | **0.6899** |
| Balance | 0.8258 | 0.8226 | **0.8323** | 0.7855 | 0.8210 | 0.8290 |
| blood | 0.6784 | 0.7662 | **0.7811** | 0.7189 | 0.7135 | 0.7716 |
| Cancer | 0.9618 | 0.9574 | 0.9574 | 0.9544 | 0.9559 | **0.9647** |
| Climate | 0.8722 | 0.9148 | 0.9148 | 0.8537 | 0.8796 | **0.9167** |
| German | 0.6550 | **0.7120** | 0.7050 | 0.6240 | 0.6460 | 0.7110 |
| LetRec | 0.7897 | 0.7143 | 0.7379 | 0.7841 | **0.7935** | 0.7476 |
| Usps | 0.8631 | 0.7758 | 0.8179 | **0.8665** | 0.8614 | 0.8222 |
| Satimage | **0.8672** | 0.8342 | 0.8566 | 0.8594 | 0.8617 | 0.8588 |
| Pendigits | 0.9630 | 0.8779 | 0.9196 | 0.9625 | **0.9678** | 0.9392 |
| Gisette | 0.8907 | 0.8372 | 0.8541 | 0.8867 | **0.8910** | 0.8730 |
| Mnist | 0.8527 | 0.7720 | 0.8055 | **0.8553** | 0.8527 | 0.8135 |
| Homus | 0.4508 | 0.4289 | 0.4481 | 0.4560 | **0.4572** | 0.4386 |
| Nist | 0.4795 | 0.4507 | 0.4684 | 0.4853 | **0.4858** | 0.4605 |
| Connect4 | 0.6222 | 0.6613 | **0.6743** | 0.6216 | 0.6197 | 0.6659 |
| Covtype | 0.9314 | 0.7752 | 0.8318 | 0.9313 | **0.9315** | 0.8533 |
| Poker | 0.5372 | 0.5881 | **0.5889** | 0.5366 | 0.5351 | 0.5870 |
| SUSY | 0.7098 | 0.7547 | **0.7651** | 0.7103 | 0.7093 | 0.7599 |
| HIGGS | 0.5860 | 0.5998 | **0.6062** | 0.5857 | 0.5856 | 0.6010 |
| Average | 0.7339 | 0.7224 | 0.7386 | 0.7294 | 0.7360 | 0.7418 |

**Table 5.** Time (ms) consumed by the proposed methods DT0–DT4 to build their DTs compared to that of the FPBST to build its BST, this is the average training time of the 10 folds.

| Dataset | FPBST | DT0 | DT1 | DT2 | DT3 | DT4 |
|---|---|---|---|---|---|---|
| Abalone | 196 | 170 | 163 | 164 | **162** | 156 |
| Australian | 45 | 44 | **37** | 39 | 39 | 42 |
| Balance | 15 | **8** | 8 | 9 | **8** | 10 |
| blood | 19 | **13** | 13 | 13 | 13 | 15 |
| Cancer | 29 | 11 | 9 | **8** | 9 | 20 |
| Climate | 29 | 21 | 21 | **20** | **20** | 30 |
| German | 56 | 62 | **61** | 68 | 62 | 70 |
| LetRec | 1595 | 1249 | 1151 | **1142** | 1176 | 1338 |
| Usps | 8096 | **8030** | 8119 | 8084 | 8098 | 9922 |
| Satimage | 860 | 743 | 769 | **771** | 774 | 958 |
| Pendigits | 910 | **574** | 576 | 578 | 581 | 882 |
| Gisette | 76,157 | 60,297 | **58,053** | 59,413 | 58,600 | 72,881 |
| Mnist | 142,712 | 118,787 | 118,678 | 117,771 | **116,226** | 141,527 |
| Homus | 83,337 | 76,153 | **72,626** | 73,118 | 73,821 | 77,758 |
| Nist | 118,746 | 99,569 | **98,802** | 100,926 | 98,987 | 106,223 |
| Connect4 | 6860 | 5832 | 5854 | 5408 | **5323** | 5857 |
| Covtype | 94,265 | 73,359 | **71,481** | 75,018 | 72,736 | 90,682 |
| Poker | 74,536 | 70,164 | 66,805 | 68,613 | **61,009** | 75,957 |
| SUSY | 923,749 | 955,593 | 948,978 | 959,964 | **906,365** | 1,017,843 |
| HIGGS | 2,974,260 | 3,117,121 | **2,855,369** | 2,958,233 | 2,951,326 | 2,939,329 |

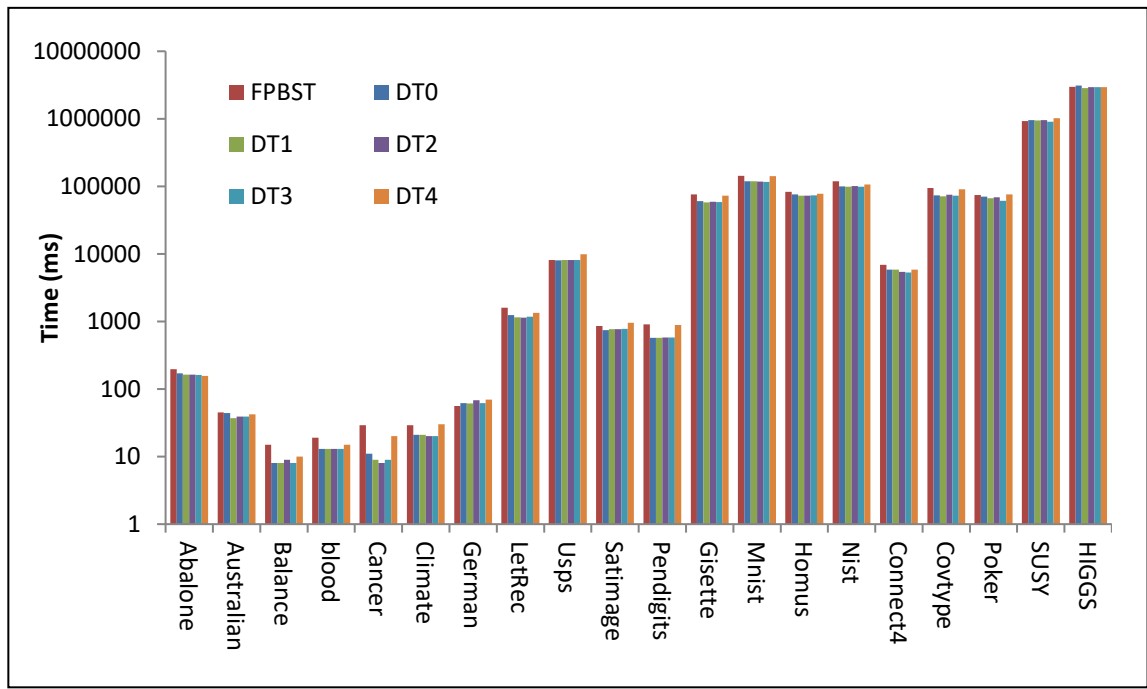

**Figure 7.** Illustration of data from Table 5 (time (ms) consumed by the proposed methods to build their DTs compared to that of the FPBST), the time axis is logarithmic base-10.

**Table 6.** Time (ms) consumed by the FPBST to test the entire test examples compared to that of the proposed methods DT0–DT4, this is the average test time of the 10 folds.

| Dataset | FPBST | DT0 | DT1 | DT2 | DT3 | DT4 |
|---|---|---|---|---|---|---|
| Abalone | 13.5 | 10.5 | 10.3 | 9.8 | 10.1 | **8.6** |
| Australian | 3.1 | 2.7 | **2.0** | 2.2 | **2.0** | **2.0** |
| Balance | 1.5 | 0.9 | **0.5** | 1.0 | **0.5** | 0.8 |
| blood | 1.9 | 1.3 | **1.0** | **1.0** | 1.1 | **1.0** |
| Cancer | 3.1 | 1.1 | **0.7** | **0.7** | 0.8 | 1.3 |
| Climate | 2.1 | 1.4 | 1.2 | **1.0** | 1.1 | 1.9 |
| German | 3.5 | **3.4** | 3.5 | 3.8 | **3.4** | 4.0 |
| LetRec | 85.2 | 69.4 | 60.2 | **59.8** | 60.9 | 67.0 |
| Usps | 390.3 | 361.4 | 365.4 | 366.3 | **361.2** | 441.9 |
| Satimage | 47.9 | 38.7 | 38.7 | **38.0** | 38.8 | 48.8 |
| Pendigits | 53.9 | 33.6 | 33.6 | **32.3** | 34.8 | 46.8 |
| Gisette | 3752 | 2920 | **2804** | 2865 | 2884 | 3480 |
| Mnist | 6869 | 5631 | 5563 | **5474** | 5571 | 6591 |
| Homus | 3965 | 3520 | 3492 | 3695 | 3540 | **3443** |
| Nist | 5686 | 5088 | 5728 | 5236 | **4893** | 5058 |
| Connect4 | 395 | **304** | 308 | **304** | 308 | 324 |
| Covtype | 4935 | **3561** | 3719 | 3662 | 3764 | 4326 |
| Poker | 4243 | 3834 | 3546 | 3641 | **3381** | 3924 |
| SUSY | 46,906 | 46,521 | 47,801 | **46,027** | 46,430 | 50,604 |
| HIGGS | 164,652 | 161,623 | **142,220** | 151,414 | 149,363 | 164,853 |

**Table 7.** Speed-up results (training and testing phases) of the proposed methods DT0–DT4 compared to the FPBST.

| Dataset | DT0 Speed | | DT1 Speed | | DT2 Speed | | DT3 Speed | | DT4 Speed | |
|---|---|---|---|---|---|---|---|---|---|---|
| | **Train** | **Test** | **Train** | **Test** | **Train** | **Test** | **Train** | **Test** | **Train** | **Test** |
| Abalone | 1.15 | 1.29 | 1.20 | 1.31 | 1.19 | 1.38 | 1.21 | 1.34 | 1.25 | 1.57 |
| Australian | 1.03 | 1.15 | 1.21 | 1.55 | 1.17 | 1.41 | 1.16 | 1.55 | 1.06 | 1.55 |
| Balance | 1.82 | 1.67 | 1.80 | 3.00 | 1.78 | 1.50 | 1.89 | 3.00 | 1.50 | 1.88 |
| blood | 1.40 | 1.46 | 1.44 | 1.90 | 1.44 | 1.90 | 1.47 | 1.73 | 1.22 | 1.90 |
| Cancer | 2.72 | 2.82 | 3.23 | 4.43 | 3.59 | 4.43 | 3.30 | 3.88 | 1.46 | 2.38 |
| Climate | 1.40 | 1.50 | 1.41 | 1.75 | 1.42 | 2.10 | 1.45 | 1.91 | 0.97 | 1.11 |
| German | 0.90 | 1.03 | 0.92 | 1.00 | 0.82 | 0.92 | 0.90 | 1.03 | 0.80 | 0.88 |
| LetRec | 1.28 | 1.23 | 1.39 | 1.42 | 1.40 | 1.42 | 1.36 | 1.40 | 1.19 | 1.27 |
| Usps | 1.01 | 1.08 | 1.00 | 1.07 | 1.00 | 1.07 | 1.00 | 1.08 | 0.82 | 0.88 |
| Satimage | 1.16 | 1.24 | 1.12 | 1.24 | 1.11 | 1.26 | 1.11 | 1.23 | 0.90 | 0.98 |
| Pendigits | 1.59 | 1.60 | 1.58 | 1.60 | 1.57 | 1.67 | 1.57 | 1.55 | 1.03 | 1.15 |
| Gisette | 1.26 | 1.28 | 1.31 | 1.34 | 1.28 | 1.31 | 1.30 | 1.30 | 1.04 | 1.08 |
| Mnist | 1.20 | 1.22 | 1.20 | 1.23 | 1.21 | 1.25 | 1.23 | 1.23 | 1.01 | 1.04 |
| Homus | 1.09 | 1.13 | 1.15 | 1.14 | 1.14 | 1.07 | 1.13 | 1.12 | 1.07 | 1.15 |
| Nist | 1.19 | 1.12 | 1.20 | 0.99 | 1.18 | 1.09 | 1.20 | 1.16 | 1.12 | 1.12 |
| Connect4 | 1.18 | 1.30 | 1.17 | 1.28 | 1.27 | 1.30 | 1.29 | 1.28 | 1.17 | 1.22 |
| Covtype | 1.28 | 1.39 | 1.32 | 1.33 | 1.26 | 1.35 | 1.30 | 1.31 | 1.04 | 1.14 |
| Poker | 1.06 | 1.11 | 1.12 | 1.20 | 1.09 | 1.17 | 1.22 | 1.25 | 0.98 | 1.08 |
| SUSY | 0.97 | 1.01 | 0.97 | 0.98 | 0.96 | 1.02 | 1.02 | 1.01 | 0.91 | 0.93 |
| HIGGS | 0.95 | 1.02 | 1.04 | 1.16 | 1.01 | 1.09 | 1.01 | 1.10 | 1.01 | 1.00 |
| Average | 1.28 | 1.33 | 1.34 | 1.55 | 1.34 | 1.48 | 1.36 | 1.52 | 1.08 | 1.27 |

As can be seen from Table 6 and Figure 8, the consumed time in the testing phase for DT0–DT3 is less than that of the FPBST, this is due to the smaller size of these threes, and the disuse of the KNN classifier, it is interesting to note that the DT4 speed in the testing phase is almost similar to that of the FPBST, this is due the large and equal size of their trees. It is also interesting to note that there is no significant difference in the time consumed by the proposed DT0–DT3 in the testing phases, since they are almost the same except for the method of calculating the probability for each class.

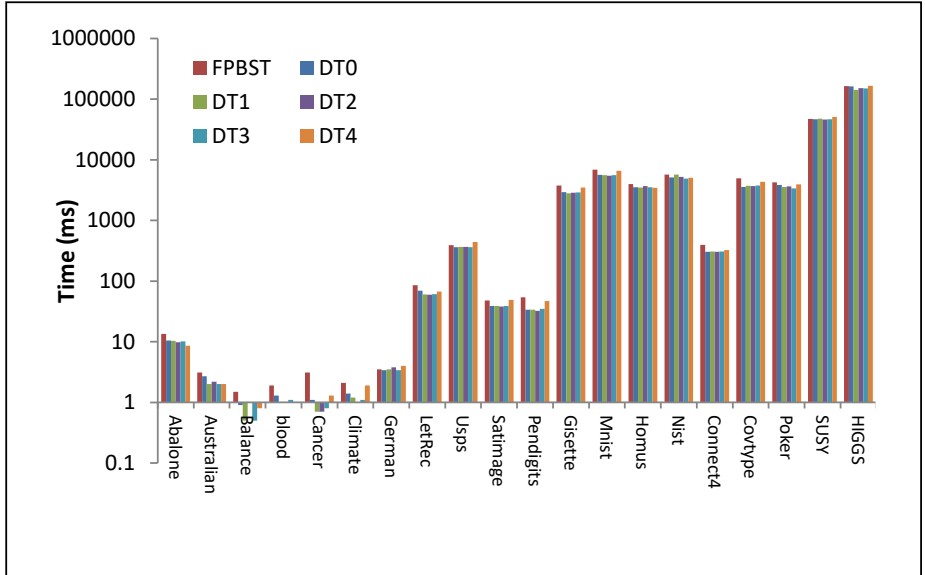

**Figure 8.** Illustration of data from Table 6 (time (ms) consumed by the FPBST to test the entire test examples compared to that of the proposed methods), the time axis is logarithmic base-10.

The speed up results shown in Table 7 and Figure 9 are calculated from both Table 5 (the speed of training phase), and Table 6 (the speed of testing phase) using Equation (7). Here, we can see that the

speed of DT4 in training phases is almost similar to that of the FPBST, this is due to the similar trees created by both methods, however the speed of the DT4 in the testing phases is significant 1.27 times of the FPBST testing phases on average, this is due to the disuse of the KNN by DT4. It is interesting to note the high speed of the proposed DT0–DT3 methods, which is about one and half times faster than the FPBST, which might be due to the smaller size of the resultant trees and the disuse of the KNN. It is also interesting to note that the training speeds of the proposed DT0–DT3 are not significant as their testing speeds; this is due to the extra time which is needed for the extra computations of the probabilities of the classes in each node.

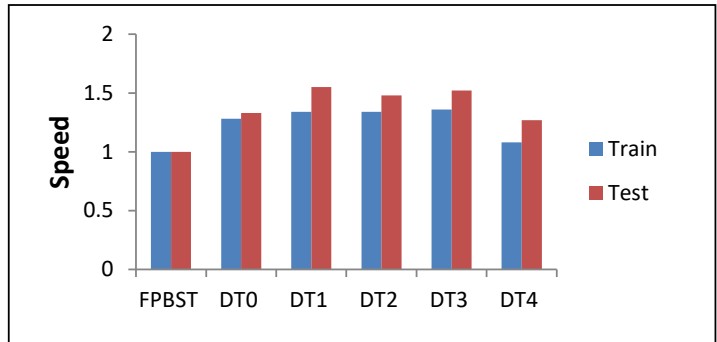

**Figure 9.** Illustration of data average from Table 7 (speed-up results—training and testing phases—of the proposed methods DT0–DT4 compared to the FPBST).

As mentioned above, the proposed DT0–DT4 have been further improved attempting to provide more regular trees in terms of class distribution, this improvement includes the enforcement of the examples that are similar to the furthest point of the lower class to be sorted to the left-side of the tree, and those which are similar to the other furthest point with the higher class to be sorted to the right-side of the tree. We conducted several experiments to measure the effect of this improvement on both accuracy and speed. Here we choose the DT of the best performance on each dataset and compare its performance to the FPBST, the comparison results are shown in Table 8.

**Table 8.** Accuracy (Acc.) and speed-up results (training and testing phases) of FPBST compared to the proposed DT0+ to DT4+.

| Dataset | FPBST | | | | DT+ | | | DT+ Speed | |
|---|---|---|---|---|---|---|---|---|---|
| | Acc. | Train | Test | DT | Acc. | Train | Test | Train | Test |
| Abalone | 0.4990 | 196 | 14 | 0+ | **0.5441** | 206 | 12 | 0.95 | 1.13 |
| Australian | 0.6435 | 45 | 3 | 4+ | **0.6957** | 48 | 3 | 0.94 | 1.03 |
| Balance | 0.8258 | 15 | 2 | 1+ | **0.8468** | 10 | 1.2 | 1.50 | 1.25 |
| blood | 0.6784 | 19 | 2 | 1+ | **0.7635** | 19 | 1.5 | 0.98 | 1.27 |
| Cancer | **0.9618** | 29 | 3 | 4+ | 0.9574 | 32 | 2.9 | 0.92 | 1.07 |
| Climate | 0.8722 | 29 | 2 | 4+ | **0.9148** | 32 | 2.1 | 0.91 | 1.00 |
| German | 0.6550 | 56 | 4 | 0+ | **0.7100** | 52 | 3.4 | 1.08 | 1.03 |
| LetRec | **0.7897** | 1595 | 85 | 3+ | 0.7884 | 1268 | 68.6 | 1.26 | 1.24 |
| Usps | 0.8631 | 8096 | 390 | 2+ | **0.8694** | 6456 | 306.3 | 1.25 | 1.27 |
| Satimage | **0.8672** | 860 | 48 | 3+ | 0.8652 | 670 | 34.8 | 1.28 | 1.38 |
| Pendigits | 0.9630 | 910 | 54 | 3+ | **0.9642** | 583 | 31.7 | 1.56 | 1.70 |
| Gisette | **0.8907** | 76,157 | 3752 | 3+ | 0.8904 | 57,101 | 2762.8 | 1.33 | 1.36 |
| Mnist | **0.8527** | 142,712 | 6869 | 2+ | 0.8523 | 118,015 | 5493.6 | 1.21 | 1.25 |
| Homus | **0.4508** | 83,337 | 4023 | 3+ | 0.4468 | 80,330 | 3554 | 1.04 | 1.13 |
| Nist | 0.4795 | 118,746 | 5686 | 3+ | **0.4828** | 108,402 | 4775 | 1.10 | 1.19 |
| Connect4 | 0.6222 | 6860 | 395 | 1+ | **0.6723** | 5735 | 298 | 1.20 | 1.33 |
| Covtype | **0.9314** | 94,265 | 4935 | 3+ | 0.9312 | 74,155 | 3728 | 1.27 | 1.32 |
| Poker | 0.5372 | 74,536 | 4243 | 1+ | **0.5889** | 66,029 | 3570 | 1.13 | 1.19 |
| SUSY | 0.7098 | 923,749 | 46,906 | 1+ | **0.7639** | 864,713 | 44,285 | 1.07 | 1.06 |
| HIGGS | 0.5860 | 2,974,260 | 164,652 | 1+ | **0.6061** | 2,854,760 | 141,130 | 1.04 | 1.17 |
| Average | 0.7339 | 225,324 | 12,103 | | 0.7577 | 211,931 | 10,503 | 1.15 | 1.22 |

As can be seen from Table 8, the accuracy of the proposed DT after the improvement has increased by about 2.38%, this is due to the sorting of the examples based on their classes, which is the only change that has been made to the decision trees. However, there is no improvement in the speed of both training and testing phases, since swapping the furthest points based on their classes need the same computation of swapping them based on their minimum/maximum norms, so there is no extra calculations needed by the new improvement, and that why the time consumed by both phases is not improved.

In order to statistically analyze the accuracy results of the DT+ methods compared to that of the FPBST (Table 8), we used the statistical test for algorithm comparison (STAC) (http://tec.citius.usc.es/stac/) [29]. Here we opt for the *t*-test paired as being commonly used to determine whether there is a significant difference between the means of two groups; the first group is FPBST results and the second group is the DT+ results. However, to do the *t*-test, our data should satisfy: independence, normality, and homocedasticity [29].

Our data is independent, since it comes from different methods (FPBST and DT+). To test the Normality of our data, we used the Shapiro-Wilk test because it performs the best, especially for samples of less than 30 elements [30]. The null hypothesis for normality would be: The samples follow a normal distribution. With significance level of 0.05, Table 9 shows the normality results.

**Table 9.** Normality test of FPBST and DT+ accuracies obtained from Table 8.

| Dataset | Statistic | *P*-Value | Result |
|---------|-----------|-----------|--------|
| FPBST | 0.93589 | 0.2003 | Null hypothesis is accepted |
| DT+ | 0.92717 | 0.13622 | Null hypothesis is accepted |

As can be seen from Table 9, according to the *p*-value of the Shapiro-Wilk test we accept the Null hypothesis, i.e., both FPBST and DT+ accuracies follow normal distributions. To test the homocedasticity of both FPBST and DT+ accuracies we opt for the Levene test [31], The null hypothesis of the homocedasticity of our data is: All the input populations come from populations with equal variances. With significance level of 0.05, Table 10 shows the homocedasticity results.

**Table 10.** Homocedasticity test of FPBST and DT+ accuracies obtained from Table 8.

| Statistic | *P*-Value | Result |
|-----------|-----------|--------|
| 0.45998 | 0.50174 | Null hypothesis is accepted |

As can be seen from Table 10, since the *p*-value is greater than the level of significance (0.05), the null hypothesis is accepted, and therefore the tested data is homocedastic. Since we verified normality, homocedasticity, and independence of our data we can apply the *t*-test. The null hypothesis (Ho): Accuracies of FPBST and DT+ have identical mean values. The alternative hypothesis (Ha): Accuracies of FPBST and DT+ do not have identical mean values. With significance level of 0.05, Table 11 shows the *t*-test results.

**Table 11.** T-test results of group 1 (FPBST) and group 2 (DT+) accuracies obtained from Table 8.

| T Statistic | *P*-Value | Result |
|-------------|-----------|--------|
| –3.8335 | 0.00112 | Ho is rejected |

As can be seen from Table 11, since the *p*-value is less than the level of significance (0.05), the null hypothesis is rejected to the favor of Ha. Thus, there is a statistically significant difference between the FPBST and the DT+ accuracies. In addition, the T statistic is shown in negative value because the mean of the FPBST accuracies is less than that of the DT+, therefore, we can say with some confidence that the DT+ methods outperform FPBST in terms of classification accuracy.

It is worth mentioning that the STAC platform accepted the Ho regardless the very small P-value, however, we rejected it here due to the p-value being less than the level of significance, which is (0.05) in our case. We further verified this decision using other platforms such as http://www.statskingdom.com/160MeanT2pair.html, in addition to our own calculations using Microsoft Excel software.

According to the previous results, we can approximately compare the performance of the proposed methods to that of the FPBST in terms of accuracy, training/testing time, and space consumed, as shown in Table 12.

**Table 12.** Approximate comparison of the performance of the proposed methods to that of the FPBST.

| Method | Training Time | Testing Time | Model Size | Accuracy |
|--------|---------------|--------------|------------|----------|
| FPBST | long | long | large | moderate |
| DT0 | short | short | small | low |
| DT1 | short | short | small | moderate |
| DT2 | short | short | small | low |
| DT3 | short | short | small | moderate |
| DT4 | long | long | large | high |
| DT0+ | short | short | small | high |
| DT1+ | short | short | small | high |
| DT2+ | short | short | small | moderate |
| DT3+ | short | short | small | high |
| DT4+ | long | long | large | high |

As can be seen from Table 12, DT0+, DT1+ and DT3+ are the best performer in general, since they obtain the highest accuracies with the shortest training/testing times and the smallest model size. And therefore, we compared them with some of the well-known decision trees, namely, J48 [32], Reduced-error pruning tree (REPTree) [33] and Random Forest (RF) [34]. We applied these trees using the Weka data mining tool [35]. The comparison results are shown in Table 13.

**Table 13.** Accuracy results of some of the proposed DTs compared to that of some of the well-known decision trees.

| Dataset | J48 | REPTree | RF | DT0+ | DT1+ | DT3+ |
|---------|-----|---------|-----|------|------|------|
| Abalone | 0.5281 | 0.5286 | 0.5430 | **0.5439** | 0.5403 | 0.5002 |
| Australian | 0.8522 | 0.8478 | **0.8754** | 0.6783 | 0.6899 | 0.6479 |
| Balance | 0.7664 | 0.7648 | 0.8176 | 0.8339 | 0.8323 | **0.8371** |
| blood | **0.7781** | 0.7741 | 0.7273 | 0.7716 | 0.7689 | 0.7297 |
| Cancer | 0.9605 | 0.9531 | **0.9707** | 0.9691 | 0.9691 | 0.9574 |
| Climate | 0.9259 | 0.9148 | **0.9370** | 0.9148 | 0.9093 | 0.8630 |
| German | 0.7390 | 0.7390 | **0.7630** | 0.7100 | 0.6990 | 0.6640 |
| LetRec | 0.8825 | 0.8424 | **0.9645** | 0.7137 | 0.7463 | 0.7943 |
| Usps | 0.8943 | 0.8780 | **0.9607** | 0.7823 | 0.8200 | 0.8626 |
| Satimage | 0.8623 | 0.8595 | **0.9197** | 0.8381 | 0.8569 | 0.8619 |
| Pendigits | 0.9637 | 0.9518 | **0.9916** | 0.8802 | 0.9181 | 0.9663 |
| Gisette * | - | - | - | - | - | - |
| Mnist * | - | - | - | - | - | - |
| Homus * | - | - | - | - | - | - |
| Nist * | - | - | - | - | - | - |
| Connect4 * | - | - | - | - | - | - |
| Covtype * | - | - | - | - | - | - |
| Poker * | - | - | - | - | - | - |
| SUSY * | - | - | - | - | - | - |
| HIGGS * | - | - | - | - | - | - |

* The well-known decision trees took a very long time when tested on the big datasets (Gisette, Mnist, Homus, Nist, Connect4, Covtype, Poker, SUSY, and HIGGS), and more problematically, the Weka software crashed due to insufficient memory error, therefore, we could not record their results on these datasets.

As can be seen from Table 13, the compared DTs show competing accuracy results compared to the well-known decision trees, with and extra advantage of the ability to work on Big Data and, therefore, can be used in practice, particularly for big data and image classification such as [36–38].

## 5. Conclusions

In this paper, we propose a new approach to improve the performance of the FPBST when classifying small, intermediate and Big Datasets. The major improvement includes the abandonment of the slow KNN, which is used with the FPBST to classify a small number of examples found in a leaf-node. Instead, we convert the BST to be a decision tree by its own, seizing the labeled examples in the training phase, by calculating the probability of each class in each node, we used various methods to calculate these probabilities.

The experimental results show that the proposed decision trees improve the performance of the FPBST in terms of classification accuracy, training speed, testing speed and the size of the model (the tree in our case). We also made another simple enhancement on the FPBST algorithm in the training phase, which is the swapping of the furthest pair of points based on their classes rather than being based on their minimum/maximum Euclidean norms. This makes the resultant decision tree more coherent in terms of the distribution of the classes, making them closer to each other as possible as could, such enhancement further improved the accuracy of the proposed decision trees compared to that of the FPBST as the results suggest.

The improvements made in this study allow the proposed DTs to be more suitable for variety of real-world applications such as image classifications, face recognition, hand biometrics, fingerprint systems, and other types of biometrics, particularly, when the training data is very large, where traditional classification methods become impractical.

However, this approach is still based on finding the furthest-pair of points (diameter), which has two major disadvantages, first, it takes time to find the diameter of the data at each node, and second, these two points might be belonging to the same class, and this might affect the classification accuracy. Therefore, we need to find a more accurate and perhaps a faster algorithm to cluster the data in each node. Moreover, the Euclidean distance might not be the perfect choice for the proposed methods, therefore, we need to investigate other distance metrics such as [39,40]. These limitations will be addressed in our future work.

**Funding:** This research received no external funding.

**Conflicts of Interest:** The author declares no conflict of interest.

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
