# Peer review of "Furthest-Pair-Based Decision Trees: Experimental Results on Big Data Classification"

_information, doi:10.3390/info9110284_

Reviewer 1 Report

The authors tried to speed up the Big Data classification using a KNN classifier.

To improve the efficiency of this kNN  classifier, Prototype Selection (PS) strategies usually used. The authors should also survey PS methods and include some of them in their experiments.

A statistical test should be used for the comparison of the examined methods.

Author Response

Dear reviewer, thank you very much for your time and these inspiring and encouraging comments that we received, we have responded to each, which made our paper more suitable for publication, here a detailed response to each comment. 

Reviewer's comments and responses:

The authors tried to speed up the Big Data classification using a KNN classifier.

To improve the efficiency of this kNN  classifier, Prototype Selection (PS) strategies usually used. The authors should also survey PS methods and include some of them in their experiments. 

response: In this work, we did not use the KNN, on the contrary, all the work is about removing the need for using the slow KNN.

A statistical test should be used for the comparison of the examined methods.

response: We added table 9, summarizing a comparison of all the proposed methods.

Reviewer 2 Report

In this paper, the authors tackle an important problem of classifying large datasets in the context of big data (where the problem is not only a big volume of the data, but also its veracity, velocity and variety). The authors aimed at improving their previous algorithm, the furthest-pair-based BST, and at removing the use of the k-NN classifier (which is very slow during the inference). The manuscript is well-written and sound, however I found some issues that need resolving:

(1) The authors did not mention the training set selection as a potential remedy to effectively handle big datasets. See the following review paper which summarizes algorithms for training support vector machines from such data (these approaches are often generic and can be generalized to other classifiers as well): https://link.springer.com/article/10.1007/s10462-017-9611-1 - it would be good to mention that and add a paragraph in the related-literature section on that.

(2) I suggest referencing pseudocodes from text (in Sect. 3.1) to make them easier to follow.

(3) It would be good to improve presentation - Tables 5-7 could be more readable when presented as plots.

(4) Although the English is quite good, I suggest performing proofreading; I found some typos, and the authors mix British and American English.

Author Response

Dear reviewer, thank you very much for your time and these inspiring and encouraging comments that we received, we have responded to each, which made our paper more suitable for publication, here a detailed response to each comment. 

In this paper, the authors tackle an important problem of classifying large datasets in the context of big data (where the problem is not only a big volume of the data, but also its veracity, velocity and variety). The authors aimed at improving their previous algorithm, the furthest-pair-based BST, and at removing the use of the k-NN classifier (which is very slow during the inference). The manuscript is well-written and sound, however I found some issues that need resolving:

(1) The authors did not mention the training set selection as a potential remedy to effectively handle big datasets. See the following review paper which summarizes algorithms for training support vector machines from such data (these approaches are often generic and can be generalized to other classifiers as well): https://link.springer.com/article/10.1007/s10462-017-9611-1 - it would be good to mention that and add a paragraph in the related-literature section on that.

Response: We mentioned the training set selection and added a paragraph on page 13 about the other methods and cited the reference provided.

(2) I suggest referencing pseudocodes from text (in Sect. 3.1) to make them easier to follow. 

Response: Done

(3) It would be good to improve presentation - Tables 5-7 could be more readable when presented as plots. 

Response: Illustration figures are added.

(4) Although the English is quite good, I suggest performing proofreading; I found some typos, and the authors mix British and American English.

Response: We have made a proofreading of the manuscript, typos and minor errors were fixed where found. Language is fixed to the American English.

Reviewer 3 Report

The work presented to me for review is interesting and valuable. The problem is raised here, that the experimental results show that the proposed decision trees improve the performance of the FPBST in terms of classification accuracy, training speed, testing speed and the size of the model. Proposed solutions are original and new and propose works to be qualified for further stages of evaluation, provided that:

1. The summary lacks a table that takes into account the advantages and disadvantages of the compared classification methods

2. There is also no information in what area of human activity the proposed classification method would be applicable.

3. There is also a lack of a drawing which would represent an improvement of the presented method in terms of: classification accuracy, speed training, testing speed and the size of the model

Author Response

Dear reviewer, thank you very much for your time and these inspiring and encouraging comments that we received, we have responded to each, which made our paper more suitable for publication, here a detailed response to each comment. 

The work presented to me for review is interesting and valuable. The problem is raised here, that the experimental results show that the proposed decision trees improve the performance of the FPBST in terms of classification accuracy, training speed, testing speed and the size of the model. Proposed solutions are original and new and propose works to be qualified for further stages of evaluation, provided that:

1.       The summary lacks a table that takes into account the advantages and disadvantages of the compared classification methods

Response: done see Table 9, and around.

2.       There is also no information in what area of human activity the proposed classification method would be applicable.

Response: We added a paragraph about the real world applications of our methods.

3.       There is also a lack of a drawing which would represent an improvement of the presented method in terms of: classification accuracy, speed training, testing speed and the size of the model

Response: we have added some plots to illustrate data in some tables.

Round  2

Reviewer 1 Report

The authors should include a well known decision tree classifier in their experiments e.g. C5.0 or CART.

A statistical test should be used for the comparison e.g. using http://tec.citius.usc.es/stac/

Author Response

1- The authors should include a well known decision tree classifier in their experiments e.g. C5.0 or CART.

Response: Done.

2- A statistical test should be used for the comparison e.g. using http://tec.citius.usc.es/stac/

Response: Done.

Round  3

Reviewer 1 Report

Accept in present form